# The Effectiveness of Liquid PUD-TiO_2_ Photocatalyst on Asphalt Pavement

**DOI:** 10.3390/ma14247805

**Published:** 2021-12-16

**Authors:** Dae-Seong Jang, Sang-Hoon Kim, Young Kim, Jae-Jun Lee, Deok-Soon An

**Affiliations:** 1Department of Civil Engineering, Jeonbuk National University, Jeonju-si 54896, Jeollabuk-do, Korea; moagala@naver.com (D.-S.J.); Beam02@naver.com (Y.K.); 295, BnD Networks, Co., Ltd., Sandanoryong-gil, Samgi-myeon, Iksan-si 54524, Jeollabuk-do, Korea; semikim77@gmail.com (S.-H.K.); lee2012@jbnu.ac.kr (J.-J.L.); 3Department of Highway & Transportation Research, Korea Institute of Civil Engineering and Building Technology, 283, Goyang-daero, Ilsanseo-gu, Goyang-si 10223, Gyeonggi-do, Korea

**Keywords:** asphalt mixture, nitrogen oxides, photocatalyst, photocatalytic reaction, PUD-TiO_2_

## Abstract

Harmful nitrogen oxides (NO_X_) are produced by vehicles, factories, mines, and power plants. In fact, over one million tons of NO_X_ are emitted into the atmosphere every year, making it the most prevalent air pollutant. Approximately 45% of the emitted NO_X_ in Korea is associated with the transportation sector. In this paper, the application of a new TiO_2_ photocatalyst on the asphalt roads to remove combustion-produced NO_X_ was studied. In an effort to overcome the known constructability, adhesion, cost, and dispersion problems associated with TiO_2_ photocatalysts, the liquid polyurethane (PUD) was added with TiO_2_ to form a mixture later known as liquid PUD-TiO_2_. Laboratory and field tests were conducted to determine the optimum amount of photocatalyst to be used and the performance of asphalt pavement coated with PUD-TiO_2_ in terms of indirect tensile strength, water susceptibility, and rutting resistance. Additionally, the performance of PUD-TiO_2_ under different humidity, wind speed, and temperature conditions was also evaluated. The results showed that the application of PUD-TiO_2_ photocatalyst on the asphalt pavements road reduces the NO_X_ available on the surface of the road. The PUD-TiO_2_ also was found to have no effects on the performance of asphalt pavement. Meanwhile, under different weather conditions, the reaction between the photocatalyst and NO_X_ is mainly affected by the humidity.

## 1. Introduction

The global rapid industrialization has triggered an increase in air pollution. Korea alongside many other countries in the world is being faced with this crisis. In Korea, according to the Ministry of the Environment, the primary sources of air pollution are manufacturing industries, followed by transportation sectors [1]. Most transportation-related pollutions come from car exhaust gases, such as carbon monoxide, hydrogen, carbon, and nitrogen oxides emitted after combustion. The number of registered cars in Korea is increasing year after year, which also means an increase in exhausted gases [2]. By June 2020, the number of vehicles registered surpassed 24 million, which was an increase of approximately 700,000 cars from the previous year. This suggests that the number of cars will keep increasing and hence increase the exhausted gases emitted. The increasing number of cars highly contributed to the increase in the amount of nitrogen oxides (NO_x_) and sulfur oxides (SO_x_) generated. On average, the amount of NO_x_ and SO_x_ emitted in Korea exceeds 1 million tons and 400,000 tons per year, respectively [3]. The nitrogen monoxide (NO) produced from the car combustion reacts with oxygen (O_2_) to form nitrogen dioxide (NO_2_), which is far more harmful to human beings than nitrogen monoxide [4,5,6].

There have been many research studies on the application of photocatalyst on the road pavements to reduce the amount of nitrogen oxides available on the surface of the road after exhaustion. Mostly in these research studies, it was found that the use of photocatalyst was effective in reducing the nitrogen oxides but there were adhesion problems between the photocatalyst and asphalt concrete, where the photocatalysts usually disappeared after the rain [7,8,9,10]. This problem slowed down the application of photocatalysts on road surfaces even though they can be very helpful in the reduction of NO_X_ produced by car combustion. Therefore, as a result, in this study, the new type of photocatalyst known as liquid polyurethane titanium oxide (PUD-TiO_2_) was developed. The liquid PUD-TiO_2_ is a combination of liquid polyurethane dispersion (PUD) (highly adhesive material) and TiO_2_ photocatalyst. An effective photocatalyst has to be photochemically active, free from corrosion, biochemically inactive, and economical to produce. The TiO_2_ photocatalyst satisfies all these criteria [11,12]. The TiO_2_ photocatalyst can be divided into two types known as anatase and rutile [13]. Anatase is the most popular type used due to its greater bandgap energy, while rutile has a greater redox. The TiO_2_ photocatalyst is characterized by photoreactivity at 380 nm in the ultraviolet wavelength. Wavelengths of 380 nm have been reported to respond effectively enough with cloudy sunlight [14,15,16].

Despite being popular and having many advantages, the adhesive properties of TiO_2_ photocatalyst have discouraged its usage on asphalt pavement and other civil engineering structures. This is due to fact that when the powdered TiO_2_ solution is applied directly, it was found that the adhesion was not so good [17,18]. The crystal nanoparticles of TiO_2_ photocatalyst are needed to be coated and fixed in a particular manner so that they will sustain in different weather conditions. In order to deal with this problem, the liquid PUD-TiO_2_ was created, which is a novel material that combines photocatalyst TiO_2_ and polyurethane dispersion (PUD) material. The PUD was mixed into the Degussa P-25 TiO_2_ photocatalyst. The mixture of PUD and TiO_2_ formed a liquid PUD-TiO_2_ photocatalyst. The polyurethane dispersion material used is composed of 70% water and 30% polyurethane. As shown in Figure 1, the positively charged liquid polyurethane particles attached to the negatively charged particles from the TiO_2_, forming micelles. After the water evaporated away, the photocatalyst hardened and remained on top of the pavement surface as shown Figure 1, ready to react with the sunlight. The micelles improved adhesion and maximized the light efficiency of the TiO_2_.

In this research, both laboratory and field tests were conducted. First, the reaction of PUD-TiO_2_ and nitrogen oxides was conducted in the controlled chamber to evaluate the performance of PUD-TiO_2_ in reducing the nitrogen oxides. Then, after successful results in the closed box conditions, the photocatalyst was applied on the surface of asphalt road pavement in the field for further evaluation. In addition, further evaluation on the performance of pavement was conducted to check if the strength and durability of the pavement were compromised after the application PUD-TiO_2_. The performance of the photocatalyst under different humidity, temperature, and wind speed was also observed.

## 2. Materials 

### 2.1. PUD-TiO_2_ Photocatalyst

In this research, liquid PUD-TiO_2_ was used to reduce the NO_x_ available on the surface of the road. The TiO_2_ photocatalyst used contained anatase and rutile phases in a ratio of about 3:1_._ The liquid PUD was mixed with a size P-25 from the Degussa company in Germany to form a liquid PUD-TiO_2_. The physical properties of the liquid PUD-TiO_2_ used are shown in Table 1 below, while Figure 2 shows the appearance of the liquid PUD-TiO_2_ photocatalyst.

### 2.2. Asphalt Mixture

The asphalt concrete mixture used in this research has a maximum aggregate size of 13 mm, and it is known as wearing course (WC-2). The aggregate gradation, aggregate gradation limits, and aggregate sieves passing percentages used are presented in Figure 3.

## 3. Methodology

### 3.1. Determination of Optimal PUD-TiO_2_ Volume

The optimum amount of photocatalyst is the amount of photocatalyst that is required to coat asphalt pavements and provides the most efficient results while at the same time being the most economical. The determination of the optimum photocatalyst amount was conducted in a closed acrylic box where the PUD-TiO_2_ and NO_2_ were put together. After photocatalytic and NO_2_ reactions, the amount of NO_x_ remaining in the air can be measured either directly or indirectly. In this research, an indirect method was used to determine the degree of photocatalyst reaction. The indirect method involves the measurement of the concentration of nitrates (the final product of the reaction) produced after the reaction. Different amounts of liquid PUD-TiO_2_ on asphalt surfaces were applied to determine the optimum amount of PUD-TiO_2_. A spray-gun was used to apply PUD-TiO_2_ on the asphalt surfaces. The spray gun used was model W77 manufactured in China. The tests were performed in a sealed acrylic box so as to create a conducive environment in which photocatalysts would react sufficiently. UV-A lamps were used to prompt photocatalyst responses. Then, nitrogen oxide (NO_2_) gas at 28 ppm was injected into an enclosed acrylic box, and after 3, 5, and 7 h, the amount of nitrate produced was measured. Reactivity was analyzed by the highest amount nitrate (${\mathrm{NO}}_{3}^{-}$) generated after the reaction. 

### 3.2. Transfusion Electron Microscope (TEM)

The smaller the size of the photocatalyst particles, the better the dispersion, and as a result, the better the photocatalytic properties [19]. TEM was used to measure the size and dispersion of TiO_2_ photocatalysts mixed with liquid PUD. The measurements were taken using a TEM device Hitachi, model SU8230 from Tokyo, Japan, and the results were analyzed from the photographs taken.

### 3.3. Strength Properties of Asphalt Mixtures

The indirect tensile strength test was conducted to determine the tensile strength of a coated asphalt mixture. Measurements were done from both coated and non-coated asphalt specimens after exposing them to an outdoor environment for a duration four months. The tensile strength ratio (TSR) tests were also conducted to evaluate the moisture susceptibility of both coated and non-coated asphalt specimens, as asphalt mixtures tend to experience a reduction in bonding strength between the asphalt binder and the aggregates when exposed to moisture, which results in the striping of the aggregates.

### 3.4. Evaluation of Adhesion Using Wheel-Tracking Test

Adhesion performance is one of the most critical factors needed to determine the effectiveness of a photocatalyst when applied to the asphalt pavement. The adhesion performance of a photocatalyst on pavement surface is often reduced when photocatalysts are subjected to friction forces exerted from wheels. The wheel pass section, the most heavily loaded road section, is severely affected by this phenomenon. In this study, the amount of photocatalyst lost in the wheel pass section was determined. The specimen was made using the Marshall compactor and then coated with the photocatalyst. The Marshall compactor used was manufactured in Gyeonggi-do, South Korea. The wheel-tracking test method was conducted to evaluate the photocatalyst adhesion behavior. The wheel-tracking method was adopted to simulate daily average traffic loads in Korea (approximately 14,000 passes), and the photocatalyst loss on the wheel pass was analyzed through Envi software (ver. 5.3).

### 3.5. Evaluation under Different Weather Conditions

Previous research evaluated the workability and efficiency of TiO_2_ as a photocatalyst built into different structures, such as sidewalks and exterior walls. The research also studied the effects that weather conditions have on the photocatalytic reactions. The effects of temperature, sunlight intensity, relative humidity, and contact time (surface area, flow rate, air flow height, etc.) were investigated. All these conditions were found to have almost insignificant effects on the workability and efficiency of TiO_2_ except for one: relative humidity. In one study, it was concluded that the photocatalyst efficiency decreases as the relative humidity increases [20]. As the flow rate and relative humidity of the air increase, the NO_X_ removal efficiency decreases. The amount of humidity in the air affects the reaction between pollutants and photocatalyst. In Belgium, where the relative humidity is high, the photocatalytic effects have been proven to be low. Relative humidity is an important factor in photocatalyst efficiency [21,22]. Since the efficiency of the photocatalyst varies depending on the amount of coating and humidity, experiments were conducted to find the most efficient combination. In this study, the effects of environmental conditions on the efficiency of the photocatalyst reaction were analyzed on an asphalt block sprayed with an optimum amount of photocatalyst. The block was kept outside to expose the photocatalysts to the external environment, and its effects on PUD-TiO_2_ were observed.

## 4. Results and Discussion

### 4.1. Optimum Application Rate

The optimal application rate of PUD-TiO_2_ photocatalyst was determined to ensure that the performance and cost requirements are satisfied. The amount of nitrate produced was measured after 3, 5, and 7 h using a spectrophotometer and nitrate kits, and the results are presented in Figure 4. When PUD-TiO_2_ was coated at a volume of 0.025 L/m^2^, the nitrate production over periods of 3 h, 5 h, and 7 h were measured at 0.55, 0.61, and 0.64 mg, and when 0.05 L/m^2^ was sprayed, the values were 0.74, 1.03, and 1.05 mg, respectively. Subsequently, the higher amount of photocatalyst had only a slight variation when compared to the 0.05 L/m^2^. Therefore, the optimum amount of spray was determined to be 0.05 L/m^2^.

### 4.2. Transfusion Electron Microscope (TEM) Analysis

The photocatalyst particles need to have a large surface area and a uniform inlet, both of which require particles that are small [23]. The dispersion of these particles must be possible even when they are in a liquid form. The TEM images of a liquid PUD-TiO_2_ mixture with a TiO_2_ photocatalyst of size P-25 from Degussa company in Germany showed that the photocatalyst particles were the same size, which confirmed that it was fully dispersed, as shown in Figure 5.

### 4.3. Strength Characteristics of the Asphalt Mixtures

Since PUD-TiO_2_ photocatalysts are proposed for use as a coating material to be applied on the surface of asphalt concrete, the strength of coated asphalt specimen was required to meet the Korean standards provided in Table 2.

The indirect tensile strength test results of the photocatalyst-coated mixtures and the non-coated mixtures were compared as shown in Figure 6. Before aging, indirect tensile strength results were at 1.05 N/mm^2^ and 1.1 N/mm^2^, respectively. Three months later, the indirect tensile strength of both mixtures had increased slightly, which was an effect that was attributed to the asphalt binder aging progress [24,25,26]. Lastly, the indirect tensile strength of the photocatalyst-coated mixture had increased to 1.67 N/mm^2^, which was slightly larger than that of the non-coated mixture, which increased to 1.63 N/mm^2^. The increased values of indirect tensile strength can be explained by the loss of internal elasticity and increase in viscosity caused by the aging of the mixture itself over time. In light of the lack of a significant difference in tensile strength that we discern in these results, it appears that the PUD-TiO_2_ photocatalyst coat had no (or a negligible) effect on tensile strength.

Tensile strength ratio (TSR) tests were conducted to determine the effects of moisture on asphalt concrete. The TSR test specimen was exposed to the outdoors conditions for four months, and tests were conducted to determine whether the liquid PUD-TiO_2_ photocatalyst has any effects. The results of TSR tests are shown in Figure 7 below.

Initially, the TSR of both the photocatalyst-coated asphalt and the non-coated asphalt mixtures had nearly the same results, with both meeting the required standards. However, after three months, the tensile strength ratio was assessed to be 0.72 for the coated mixture and 0.73 for the non-coated mixture. The notable decrease in the tensile strength ratio occurred between the third and fourth months where the results were 0.59 for the coated asphalt and 0.53 for the non-coated asphalt. The reduction in TSR values was attributed to the significant amount of rain, high temperatures, and humidity over the course of August to September, which is the summer in Korea [27,28]. However, a decreasing trend was observable in both samples, regardless of whether the photocatalyst was present, leading to the conclusion that the photocatalyst coating did not have any effects on the TSR results.

### 4.4. PUD-TiO_2_ Adhesion Evaluation Test

Image analysis of the images obtained after the TEM measurement was performed with Envi. ENVI depicts photocatalyst loss and the presence of residuals as gradations of color. The traffic volume parameter was set based on the national daily average traffic volume. Figure 8 and Figure 9 present the results of simulated daily traffic volume for 10 days and 20 days. The daily average traffic volumes were passed on the coated sample under the same temperature conditions for 10 days. The coating peeled off the center and both edges of the wheel. The green portion, which represents the lost portion, was made up of approximately 16,801 pixels out of a total of 118,780 pixels, or a 14.14% loss.

When the traffic volume was simulated for 20 days, the number of photocatalysts lost (light green, red) was 53,524 pixels out of a total of 95,427 pixels or approximately 56.09%.

### 4.5. Evaluation under Different Weather Conditions

The effects of different weather and environmental conditions on the photocatalyst reaction were evaluated. The correlations of the reaction and the presence of fine dust, humidity, nitrogen dioxide exposure, and sunlight time were measured. The photocatalyst reacts at ultraviolet wavelengths (388 nm), which is within the sunlight wavelength range, and the effectiveness of the photocatalytic reaction was measured according to the amount of nitrate produced. Figure 10 presents the amount of nitrate produced as a function of both the amount of sunlight time and humidity. In the figure below, each dot represents the average data measured in a single day for four months, and “sunlight time” refers to a time during which sunlight was reflected off the asphalt. Based on the test results, nitrate production increased as the sunlight time increased and humidity decreased, as presented in Figure 10 below. The part circled in Figure 10 shows that nitrate continued to be produced even when there was no exposure to sunlight (the sun is blocked by clouds), which suggests that sunlight time did not directly affect the photocatalytic reaction, which makes sense, since UV waves continue to reach the earth’s surface even when the sun is blocked by clouds.

Korea experiences all four seasons; in spring and autumn, the air is dry, while in summer, the temperature is hot and very humid, and in winter, the air is cold and dry. The performance of photocatalysts under different weather, i.e., temperature and humidity conditions, must be considered. Figure 11 shows the results of the amount of nitrate produced under the influence of different atmospheric temperature and humidity. Over the course of the four months during which the material was tested, the daily average temperature was between 15 and 33 °C. Considering the circled part on the left, it can be seen that results were distributed in various ways under the influence of humidity regardless of the daily temperature. The circled part on the right shows that nitrate production was measured as being more than twice the average amount of nitrate produced, the reason being that on these particular days, the fine dust level was very bad. The result showed that the atmospheric temperature was found to have no effect on the production of nitrate. According to the analysis using distribution results as a trend line, at 60% humidity, the reaction was more effective.

Wind speed closely affects ambient air conditions, including the air temperature and reactivity of the light flux. Figure 12 shows the result analysis of nitrate production according to wind speed and humidity. The average wind speed over the testing period was most often between 1.5 and 2.5 m/s, though in some rare cases, the wind speed exceeded 3 m/s. In all cases, nitrate production was measured to be less than 1 mg/L, as shown on a small circle on the left. Nitrate production increased slightly as the wind speed increased, although the difference was so small that it was difficult to determine what role wind speed played in this outcome. The average wind speed from the day when the highest rate of nitrate production was recorded was approximately 2.5 m/s. In contrast, the humidity trend line graph shows that nitrate production steadily increased as humidity approached 60%, suggesting that humidity played a much stronger role in the photocatalytic reaction than wind speed.

### 4.6. Water Quality Standards According to Nitrates Production 

The nitrates produced as the by-product of the PUD-TiO_2_ photocatalyst are dissolved in rain water and eventually make their way into groundwater. Groundwater must meet the water quality standards to ensure its usage for other purposes. The amount of nitrate produced in the asphalt surface layer was measured daily for 3 months after coating, and the average value was 0.90 mg/L. For agricultural, industrial, and domestic purposes, nitrate ions should comprise no more than 20 mg of the groundwater. Drinking water requirements are even more strict with respect to ammonia nitrogen, which must be present in quantities of 0.5 mg/L or less. Unfortunately, the nitrate ions produced after photocatalyst reaction were higher than the required standard for human usage, as shown in Table 3. Additional material will be required to ensure that the amount of nitrate ions is reduced to 0.5 mg/L or less for human uses. 

## 5. On-Site Evaluation PUD-TiO_2_-Coated Asphalt Mixture

The photocatalyst was applied on the field site to measure the efficiency of photocatalyst on removing nitrogen oxides in a real-world circumstance. Samples were required to be taken from the photocatalyst-coated pavement sections as well as of the non-coated sections of the road [29,30]. The field site experiment was conducted on Gangnam-daero road around Yangjae Station in Seoul city Korea, where the photocatalyst was uniformly coated on the road through spraying. The amount of nitrate produced in a period of 42 days was measured. The results of the amount of nitrate produced on coated and non-coated sections were measured, as shown in Table 4.

The humidity in Seoul on the day of measurement was 62%, which was high enough that it may have reduced the efficiency of the reaction.

## 6. Conclusions

The tests to determine the optimal amount of TiO_2_ photocatalyst to be applied to asphalt—indirect tensile strength, tensile strength ratio, and wheel-tracking tests—were conducted. In addition, changes in the reactivity of our photocatalyst occurring as a result of different weather and environmental conditions were also measured. The following conclusions were reached after the analysis of the result: The optimum spray amount was determined to be 0.05 L/m^2^. For the higher amount of photocatalyst, the production of nitrate, which is the product of the photocatalyst and NO_X_, was of the same amount as that of 0.05 L/m^2^.The strength characteristics of the coated and non-coated asphalt was assessed after being exposed to the outdoor conditions for approximately 3 months. The indirect tensile strength increased in both the photocatalyst-coated mixture and the non-coated mixture.Both photocatalyst-coated and non-coated samples saw reduced TSR values over time. The reason behind this phenomenon was believed to be the exposure to rain, high temperatures, and humid air conditions associated during the summer season.Image analysis of the PUD-TiO_2_-coated asphalt was performed after simulating the average daily traffic volume for 10 and 20 days; the results were 14.14% and 56.09% of the photocatalyst lost, respectively. Although the adhesion performance of PUD-TiO_2_ was considered to be good, additional research on adhesion performance is needed to be done to ensure the usage of this type of photocatalyst on roads.The efficiency of the liquid PUD-TiO_2_ photocatalyst was not significantly affected by sunlight time, temperature, and windspeed. For higher temperatures and faster wind speeds, the photocatalyst reactions were of greater efficiency. However, the factor that mostly influenced the photocatalyst reactions’ efficiency was humidity, with the photocatalytic reaction most effectively occurring when humidity was near 60%.

## Figures and Tables

**Figure 1 materials-14-07805-f001:**
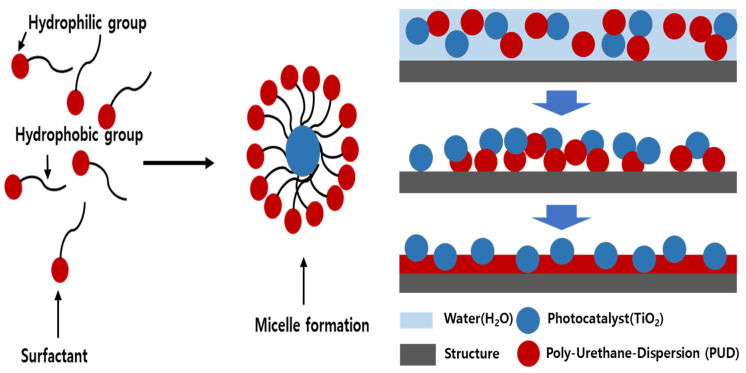
Micelle formation and hardening processes.

**Figure 2 materials-14-07805-f002:**
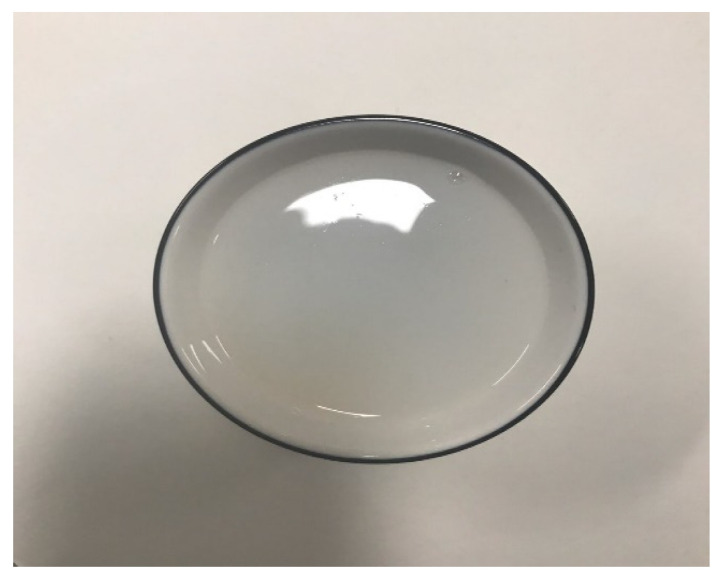
PUD-TiO_2_’s appearance.

**Figure 3 materials-14-07805-f003:**
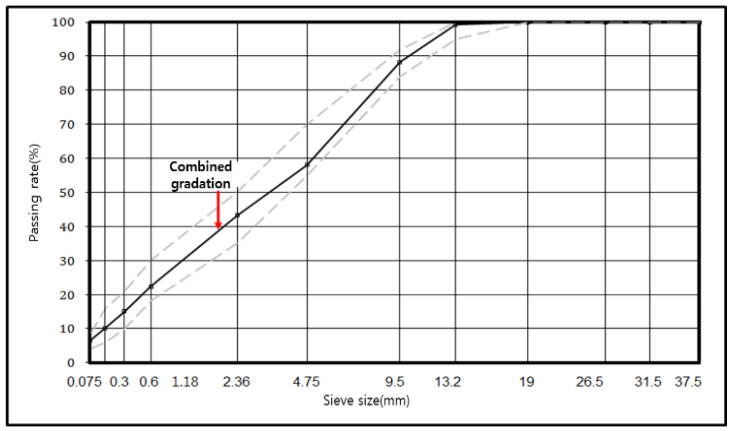
Aggregate gradation.

**Figure 4 materials-14-07805-f004:**
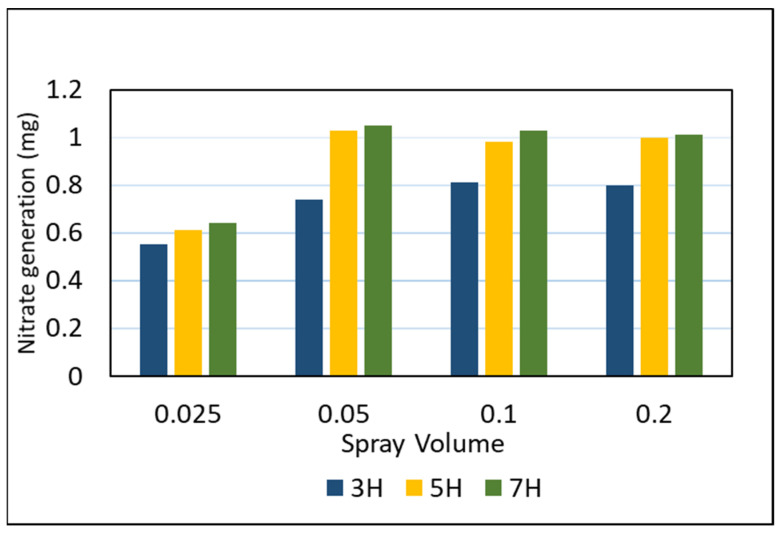
Optimal amount of photocatalyst.

**Figure 5 materials-14-07805-f005:**
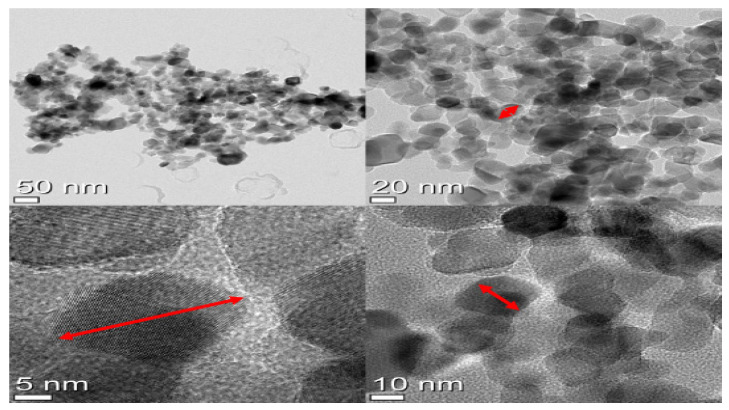
Microstructure of PUD-TiO_2_ liquid at different resolutions.

**Figure 6 materials-14-07805-f006:**
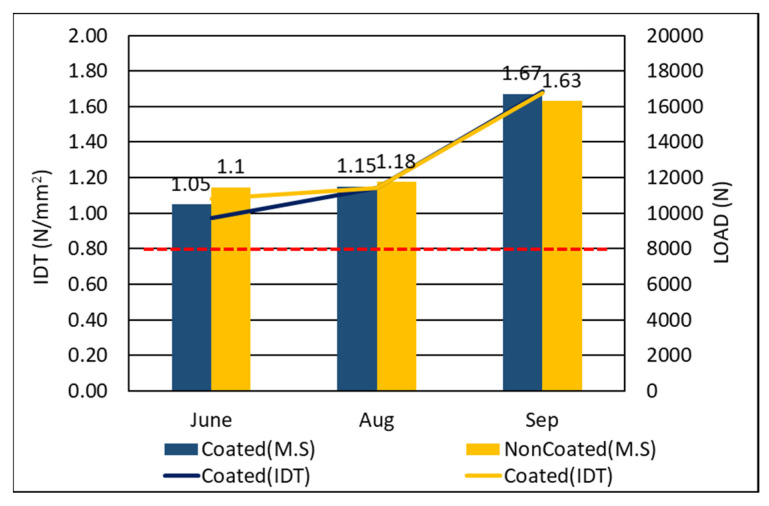
Indirect tensile strength.

**Figure 7 materials-14-07805-f007:**
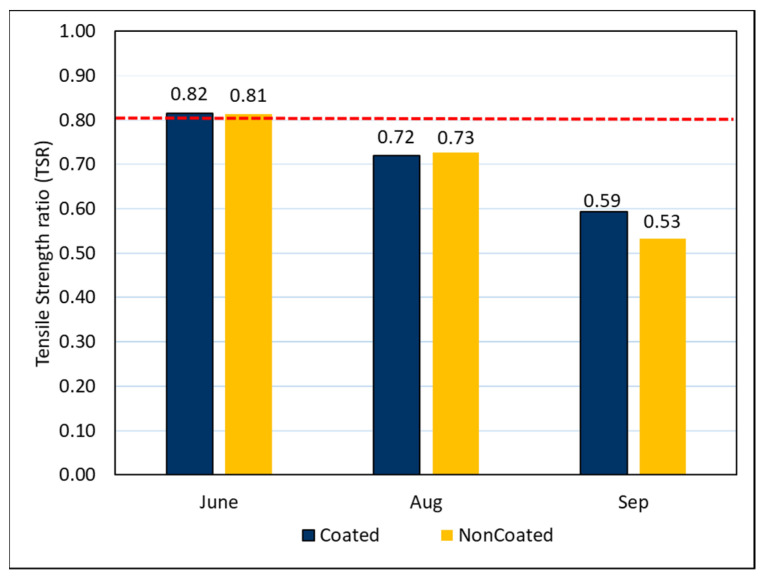
Tensile strength ratios.

**Figure 8 materials-14-07805-f008:**
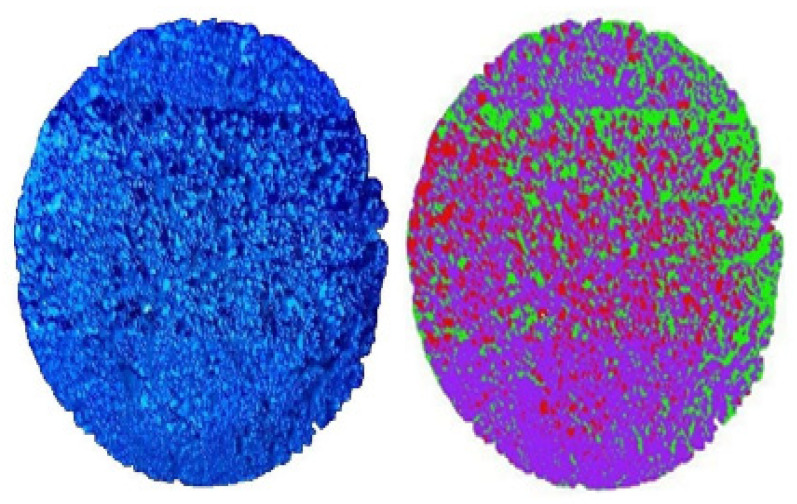
Daily traffic volume at 10 days.

**Figure 9 materials-14-07805-f009:**
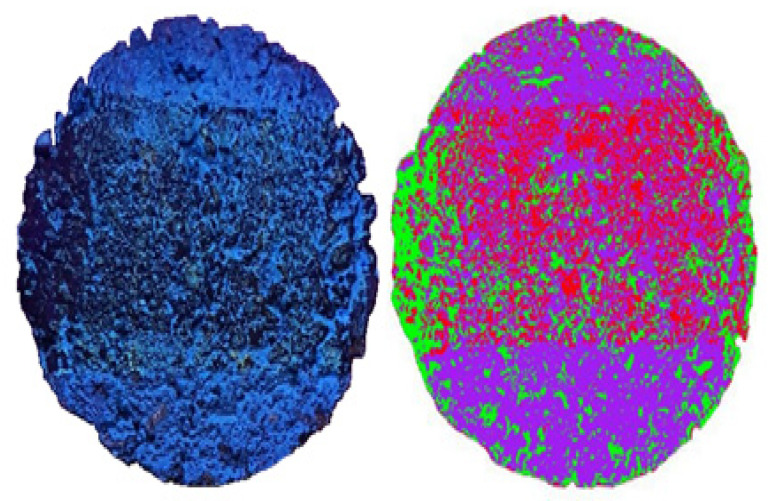
Daily traffic volume at 20 days.

**Figure 10 materials-14-07805-f010:**
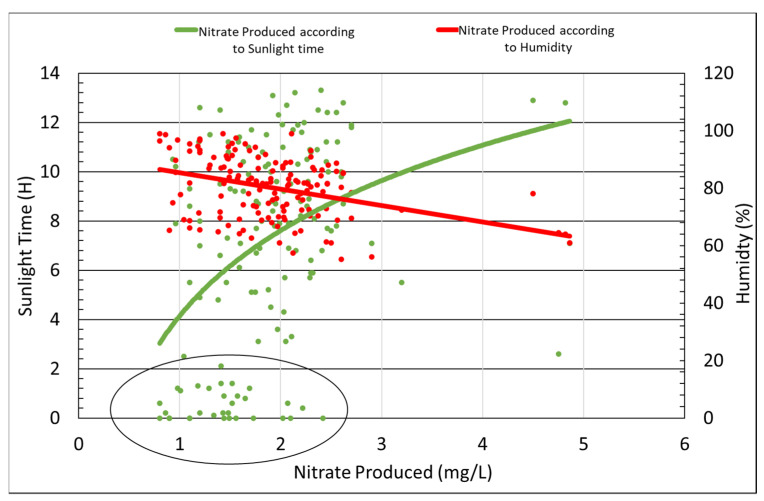
Effects of sunlight time on nitrate production.

**Figure 11 materials-14-07805-f011:**
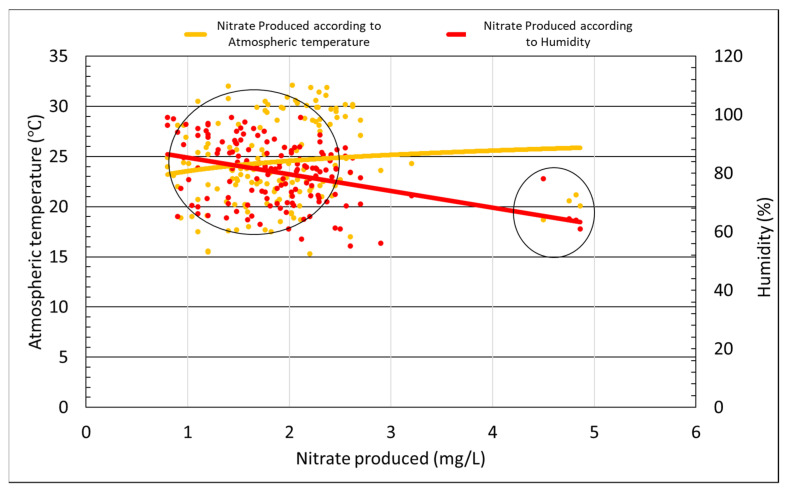
Effects of atmospheric temperature on nitrate production.

**Figure 12 materials-14-07805-f012:**
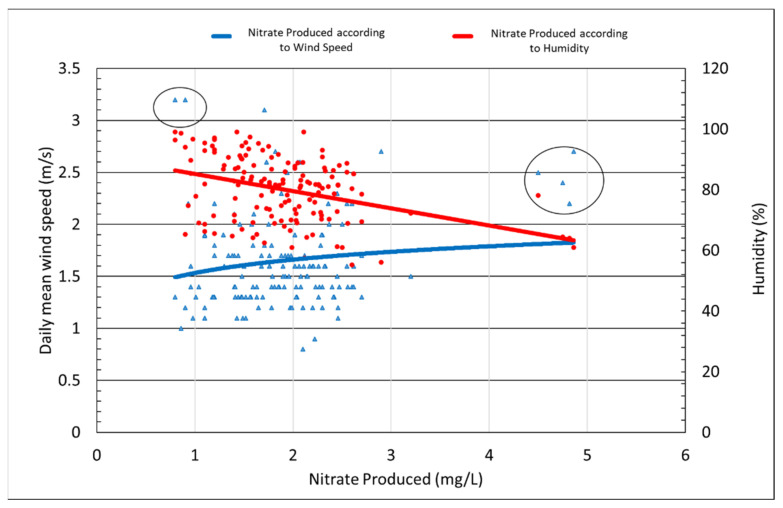
Effects of wind speed on nitrate production.

**Table 1 materials-14-07805-t001:** Physical properties of PUD-TiO_2_.

Specification	Unit	Value
Specific surface area (BET)	m^2^/g	50 ± 15
Average primary particle size	nm	21
Moisture (2 h at 105)	wt %	≤1.5
pH value	-	3.5–4.5
Titanium dioxide	wt %	≥99.50
Al_2_O_3_ content	wt %	≤0.300
SiO_2_ content	wt %	≤0.200
Fe_2_O_3_ content	wt %	≤0.010
HCl content	wt %	≤0.300
Sieve residue	wt %	≤0.050

**Table 2 materials-14-07805-t002:** The strength properties of various asphalt mixtures.

Test	WC-1~4	WC-5~6	etc.
Marshall stability (N)	Over 7500 (Over 5000)	Over 6000	Compaction of number both sides 75
Flow (1/100)	20~40		
Void (%)	3–6	3–5	
Saturation degree (%)	65–80	70–85	
TSR	Over 0.8		Void (%) 7 ± 0.5
IDT (N/mm^2^)	Over 0.8	Over 0.6	
Toughness (N.mm)	Over 8000	Over 6000	

**Table 3 materials-14-07805-t003:** Drinking water standards.

Measurement Item	Standard
Bacteria	100 CFU/mL and less
Ammonia nitrogen	0.5 mg/L and less
Nitric nitrogen	10 mg/L and less
Cd	0.06 /L and less

**Table 4 materials-14-07805-t004:** Nitrate production results.

Site Samples	Average Result (mg/L)
Coated sample #1	2.13
Coated sample #2	2.07
Non-coated sample #3	1.54

## Data Availability

Not applicable.

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
