# Peer review of "The Effectiveness of Liquid PUD-TiO_2_ Photocatalyst on Asphalt Pavement"

_materials, 2021, doi:10.3390/ma14247805_

Round 1

Reviewer 1 Report

The paper study an important problem for a developed country. Nitrogen oxides (NOX) are produced by vehicles, factories, mines, and power plants. In fact, over one million tons of NOX are emitted into the atmosphere every year, making it the most prevalent air pollutant. Approximately 45% of emitted NOX is associated with the transportation sector. In the paper is tested the capacity of a new TiO2 photocatalyst applied to asphalt roads to remove combustion-produced NOX.

Generally speaking, the manuscript is well written, the material is judiciously divided and organized and correct from scientific point of view. The scientific novelty is reduced but the practical importance is significantly. Some changes are, however, necessary. For these reasons I can recommend the acceptance of this paper after some minor corrections.

Please respect the template of the paper (check again). Please to emphasize more clearly the contribution of the manuscript from a scientific point of view. 

Author Response

Thank you for your comment. Corrections have been made.

Reviewer 2 Report

In this manuscript, the authors study the effectiveness of liquid PUD-TiO2 Photocatalyst on Asphalt Pavement. This work sounds very interesting and meaningful, and the analysis is reasonably clear, but there is still a few concerns regarding this article, therefore, I suggest it for publication in this journal after the following major revisions:

  1. In the results section, it should be compared with the existing results to explain the advantages of this paper.
  2. Mentioned in section 3.1, the amount of NOx remaining in the air can be measured directly, Can you explain why indirect measurement was chosen over direct measurement?
  3. Some grammatical errors are found. A proof-reading of the article will enhance the quality of the paper.
  4. The size and position of the picture should be optimized for placement.

Author Response

Thank you for your comment. Please find attached file.

Reviewer 3 Report

The authors of the study entitled: "Effectiveness of the liquid PUD-TiO2 photocatalyst on asphalt pavement" proposed the use of TiO2 to reduce the concentration of nitrogen oxides released during road traffic.  

In the Conclusions, they stated that the time spent in the sun did not significantly affect the efficiency of the new photocatalyst. And rightly so. How can a photocatalyst dispersed in a black mass of asphalt work? The authors should consider the sense of the conducted research.

This article does not meet the requirements of a research article. Maybe it could publish in some daily newspaper.

  • Instead of writing how many cars per inhabitant of South Korea, the authors should clearly write what for they wanted to add TiO2 to the asphalt;

- The article is sloppy written.

Instead of posting infantile pictures (Figs. 7, 9, 19), they should focus on interpreting the others. Although I have many reservations here, for example what are the numbers on the vertical axis of Fig.1; where is the comment for Fig. 2 ?; the colors in the legend in Fig. 16 are different than in the picture.

- How they examined Marshall stability (Table 2) and whether this value was of any importance for further research.

- Where did the value of 0.90 mg / L concerning the amount of nitrogen produced by the photocatalyst come from? (line 286).

- the English of this work is unacceptable.

Concluding, I am sure that the manuscript should be rejected.

Author Response

(The authors gave the same response as above.)

Reviewer 4 Report

This research tested the capacity of a new TiO2 photocatalyst applied to asphalt 20 roads to remove combustion-produced NOX. The overall work is good. The paper structure is acceptable. There is a variety of useful results. However, the following comments should be considered:

  • The introduction is too short and should be extended. Talk more in detail about the most recent studies conducted on liquid PUD-TiO2 Photocatalyst.
  • Don’t use personal pronouns (I, we, ..etc) in the paper.
  • It is better for the quality of your work to compare your findings with previous studies’ findings and to comment on these comparisons.
  • Rearrange your conclusions in terms of points, and include some recommendations for future work and for practicing engineers in the construction field.
  • Double-check the spelling and grammar.

Author Response

(The authors gave the same response as above.)

Round 2

Reviewer 2 Report

In this manuscript, the authors study the effectiveness of liquid PUD-TiO2 Photocatalyst on Asphalt Pavement. This work sounds very interesting and meaningful, and the analysis is reasonably clear, but there is still a few concerns regarding this article, therefore, I suggest it for publication in this journal after the following major revisions:

  1. The analysis of Figures 12, 13, and 14 should be a little clearer and more detailed. For example, what are the dots on the graph? The trend of the curve shows that sunshine duration is related to nitrate production. Why does the paper say sunshine duration is irrelevant?  The same problem analysis is required for the other two diagrams.
  2. It was concluded that catalytic efficiency was highest at humidity below 60%, but no tests below 60% were observed at the time of the experiment.
  3. Figure 5 seems pointless.

Reviewer 3 Report

In its present form, the article is slightly better, but it does not deserve to be published in a prestigious journal anyway 

Reviewer 4 Report

I recommend this research for publication in Materials journal

Author Response

Thank you for your recommendation.

Round 3

Reviewer 2 Report

The authors have been revised as required to meet the journal acceptance criteria.
